# Crystal structure of an adenovirus virus-associated RNA

Iris V. Hood[1], Jackson M. Gordon [1], Charles Bou-Nader[1], Frances E. Henderson[1], Soheila Bahmanjah[1] & Jinwei Zhang [1]

Adenovirus Virus-Associated (VA) RNAs are the first discovered viral noncoding RNAs. By mimicking double-stranded RNAs (dsRNAs), the exceptionally abundant, multifunctional VA RNAs sabotage host machineries that sense, transport, process, or edit dsRNAs. How VA-I suppresses PKR activation despite its strong dsRNA character, and inhibits the crucial anti-viral kinase to promote viral translation, remains largely unknown. Here, we report a 2.7 Å crystal structure of VA-I RNA. The acutely bent VA-I features an unusually structured apical loop, a wobble-enriched, coaxially stacked apical and tetra-stems necessary and sufficient for PKR inhibition, and a central domain pseudoknot that resembles codon-anticodon interactions and prevents PKR activation by VA-I. These global and local structural features collectively define VA-I as an archetypal PKR inhibitor made of RNA. The study provides molecular insights into how viruses circumnavigate cellular rules of self vs non-self RNAs to not only escape, but further compromise host innate immunity.

---

[1] Laboratory of Molecular Biology, National Institute of Diabetes and Digestive and Kidney Diseases, 50 South Drive, Room 4503, Bethesda, MD 20892, USA. Correspondence and requests for materials should be addressed to J.Z. (email: jinwei.zhang@nih.gov)

Adenoviruses infect the respiratory system and the gastrointestinal and urinary tracts and can be life-threatening for immunocompromised patients. Late in infection, adenoviruses produce two extraordinarily abundant ($10^7$–$10^8$ copies per cell), highly structured noncoding RNAs termed virus-associated RNAs (VA-I and VA-II; ~160 nucleotides (nts)), leading to their discovery as the first viral noncoding RNAs[1,2]. At least one VA RNA exists in all known adenovirus serotypes and most (~80%) contain both[3]. The multifunctional VA RNAs interfere with essentially all host systems that interface with double-stranded RNAs (dsRNAs), from their sensing by protein kinase R (PKR), export by Exportin-5, processing by Dicer, editing by ADAR, to activation of oligoadenylate synthetases (OAS), etc[4]. Dicer-processed terminal strands of VA are further assembled into functional RISC complexes that may target additional host systems[5,6]. Collectively, VA RNAs contribute ~60 fold to viral titers and confer adenoviruses general resistance to interferon-mediated antiviral defense[7].

PKR is a central component of the interferon response and the best-characterized target of VA RNAs. PKR recognizes dsRNAs produced during viral replication or bidirectional transcription, and exerts antiviral and antiproliferative effects through signaling pathways including NF-kB, TNF, STATs, p53, etc[8]. Beyond its immune function, PKR plays additional roles in neurodegenerative diseases[9], cognition and memory[10], malignant transformation[11], etc. Dimerization of PKR on sufficiently long dsRNA (>30–33 base pairs) induces PKR autophosphorylation and activation of its latent kinase activity[12]. Activated PKR phosphorylates the alpha subunit of eukaryotic translation initiation factor 2 (eIF2α) to block cap-dependent translation initiation and production of new viral particles[13]. Indeed, PKR-deficient mice are exceptionally susceptible to intranasal infection by vesicular stomatitis and influenza viruses[14]. To counter the debilitating restriction by PKR, nearly all known viruses have evolved protein or RNA antagonists to evade or inactivate PKR. The constant tug-of-war with viral antagonists has driven intense positive selection of PKR in vertebrates[15]. Besides manipulation by viral elements, PKR activity is subject to tight control by host protein factors including NF90[16], TRBP[17], ADAR1[18], etc. Furthermore, endogenous, highly structured RNAs also modulate PKR activity, such as snoRNAs[19], the IFN-γ and TNF-α pre-mRNAs[20,21], nc886 RNA[22], etc. Recently, additional endogenous RNAs have been shown to directly associate with PKR, including nuclear transposable elements and mitochondrial dsRNAs proposed to mediate mitochondria–cytosol communication, especially during stress[23].

Among the expanding collection of PKR-regulatory RNAs, VA-I is the best-characterized RNA antagonist[24]. It is comprised of an elongated apical stem loop, a central domain proposed to contain a pseudoknot, and a terminal stem thought to be dispensable for PKR binding and inhibition (Fig. 1a, b; Supplementary Fig. 1)[3,25,26]. Although the importance of higher-order RNA structure to PKR modulation is well recognized[27], no high-resolution structure of VA, or of any other complex RNA modulator of PKR is known. Nonetheless, extensive phylogenetic, biochemical, and small-angle X-ray scattering (SAXS) analyses of VA-I uncovered a number of unusual features of this RNA. The apical domain is exceptionally thermostable ($T_m$~83 and ~95 °C, in the absence and presence of 1 mM $Mg^{2+}$, respectively[26,28]), causing it to migrate much slower on denaturing Urea-PAGE and to resist denaturation by up to 6 M urea. Yet, structural-probing analyses revealed sensitivity of this region to both ssRNA-specific and dsRNA-specific RNases (RNase A and V1, respectively)[25,29]. This led to the proposal that the VA-I apical region exists as two structurally different, functionally distinct conformers[25,29]. However, it is perplexing how a single RNA region is both

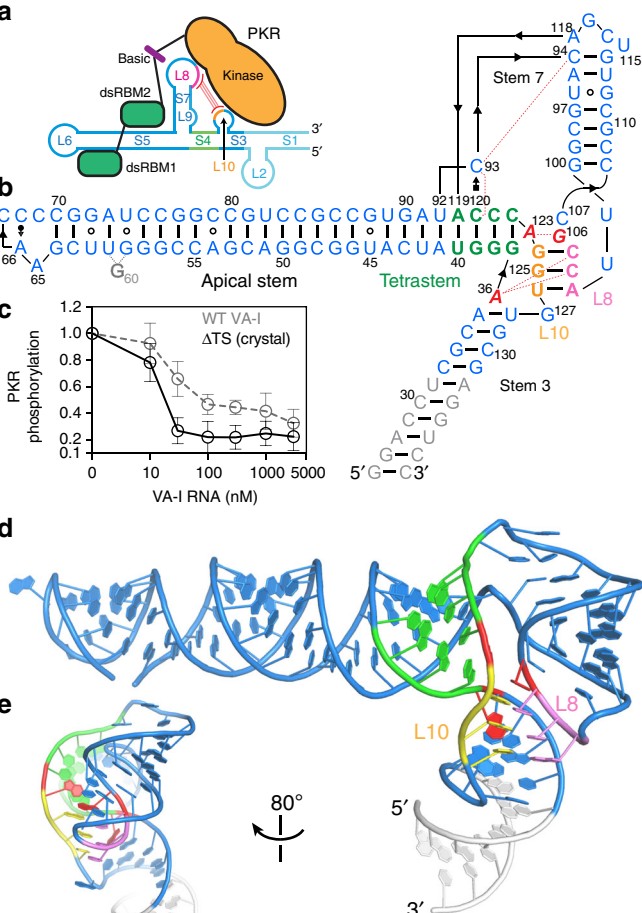

**Fig. 1** Overall structure of Adenovirus 2 VA-I RNA. **a** Cartoon model of VA-I and PKR interactions. S: stem; L: loop. S1 and L2 were absent from the crystal construct. **b** Secondary structure of crystallized VA-I RNA. Tetrastem: green, Loop 8: violet, Loop 10: orange, engineered sequences: gray. Three single-stranded, helix-capping purines, A36, G106, and A123, are in red italics. Leontis–Westhof symbols[74] denote non-canonical base pairs. Arrowheads denote chain connectivity. Dotted red lines indicate hydrogen bonds. G60 is disordered. **c** Inhibitory activities of wild-type and ΔTS (deletion of terminal stem; crystallization construct) VA-I. Several ΔTS RNAs showed an enhanced inhibitory activity than WT, as previously reported[26]. Error bars here and thereafter represent standard deviations (s.d.) from three independent replicates. **d**, **e** Two views of the overall structure of VA-I, colored as in **b**

extraordinarily stable and yet fluid enough to diverge into two distinct conformations. The structurally complex central domain has been long proposed to contain a 3-bp pseudoknot between Loop 8 and Loop 10, based on covariation of the loop sequences[25]. Yet its functional importance for PKR regulation has remained unclear. UV-melting analysis of the central domain revealed an unusual sensitivity to solution pH, the structural basis of which remains unknown[30]. At the junction between the apical and central domains is an essentially invariant tetrastem structure (Fig. 1b). The specific sequence of the tetrastem, in addition to its secondary structure, is required for full inhibitory activity against PKR[31]. What drives the near-universal sequence conservation of this functionally crucial element?

To address these open questions and understand how viral RNAs employ unique tertiary structures to confound and defeat host immune proteins, we determined a 2.7 Å crystal structure of the apical and central domains of VA-I RNA of adenovirus serotype 2 (Fig. 1b–e; Table 1; Supplementary Figs. 2 and 3). This

**Table 1 Summary of crystallographic statistics of a native dataset**

| | Native |
|---|---|
| *Data collection* | |
| Space group | $P4_122$ |
| *Cell dimensions* | |
| $a, b, c$ (Å) | 100.5, 100.5, 130.7 |
| $\alpha, \beta, \gamma$ (°) | 90, 90, 90 |
| Resolution (Å)[a] | 100.0–2.74 (2.81–2.74) |
| $R_{merge}$ (%)[a] | 15.2 (364) |
| $<I>/<\sigma(I)>$[a] | 12.2 (0.82) |
| Completeness (%)[a] | 99.9 (100.0) |
| Redundancy[a] | 8.0 (8.2) |
| $CC_{1/2}$ | 0.998 (0.340) |
| $CC^*$ | 1.000 (0.712) |
| *Refinement* | |
| Resolution (Å)[a] | 37.02–2.74 (2.84–2.74) |
| No. of reflections[b] | 18,173 (1771) |
| $R_{work}/R_{free}$ (%)[a] | 22.0 (43.0)/23.6 (42.5) |
| No. of atoms | 2362 |
| RNA | 2352 |
| Ion | 1 ($K^+$) |
| Water | 9 |
| Mean $B$-factors (Å$^2$) | 80.6 |
| RNA | 80.7 |
| Ligand/ion | 88.3 |
| Water | 63.6 |
| R.m.s. deviations | |
| Bond lengths (Å) | 0.002 |
| Bond angles (º) | 0.42 |
| Maximum-likelihood coordinate precision (Å) | 0.47 |
| PDB accession code | 6OL3 |

[a]Values in parentheses are for the highest resolution shell
[b]Values in parentheses are for the cross-validation set

RNA is derived from the Dicer-processed form and contains all the elements necessary for PKR inhibition (Fig. 1c; ΔTS for deletion of terminal stem)[26]. The structure uncovers a set of global and local features that collectively define VA-I as a potent PKR inhibitor, and gives insights into how this viral decoy of dsRNA escapes the activation of PKR and OAS1. Further, we performed structure-guided mutational analyses of VA-I to functionally demarcate different VA-I domains and interpreted previous biochemical findings in the context of the 3D structure. Finally, the structure of VA-I revealed unexpected similarities to the tRNA anticodon stem loop (ASL) and codon–anticodon interactions. Such resemblances shed light on the evolutionary origin of VA RNAs and may hint at undescribed functions of VA-I's central domain.

## Results

### Overall structure of VA-I RNA.
The VA-I structure assumes a sharply bent "V" shape, in which two arms of coaxially stacked helices converge at a stable central domain (Fig. 1b–e; Supplementary Fig. 4a–d; Supplementary Movie 1). The crystal structure establishes a secondary structure scheme that is distinct from the Rfam model (family RF00102[http://rfam.org/family/RF00102]) based on multiple sequence alignment. It also differs significantly from recently reported structural models derived from SAXS analyses and computer modeling both globally and locally[32,33]. The SAXS envelopes of VA-I generally showed obtuse angles between the apical and terminal stems, while the crystal structure captures VA-I in an acutely bent "V" shape (~60° between the two arms; Fig. 1d). This sharp bend in the dsRNA trajectory is appropriate for its PKR-inhibitory function, as it deviates more

from the colinear, or *cis* global dsRNA arrangements that generally activate PKR[34,35]. It was shown that bending a 51-bp dsRNA by inserting adenosine bulges of increasing sizes at its center progressively reduced its potential to activate PKR, by as much as 10-fold when a ~93° bend is produced with a 6-adenosine bulge[35]. Further, when two bends were introduced in the same dsRNA, a more bent *cis*-bulged global shape activated PKR far less than a more linear *trans*-bulged global shape. Contrary to the acutely bent VA-I that functions as a PKR inhibitor, the structural model of the PKR-activating TNF-α pre-mRNA 3' UTR features two parallel helices that drive exceptional PKR activation[21]. Similar near-parallel, adjacent helices in riboswitches, and ribozymes also drive PKR activation[36]. Thus, the sharply bent global shape of VA-I appears to have evolved to avoid near-linear, extended configurations that are hallmarks of PKR-activating RNAs.

### An apical structure with both ssRNA and dsRNA characteristics.
In contrast to a proposed 8-nt flexible loop, the apical region of VA-I is highly structured and heavily stacked, explaining its extraordinary thermostability, aberrant gel mobility, and resistance to denaturation by 6 M urea (Fig. 2a)[28,31]. A stack of two G–C pairs and a G·U wobble pair form the apical core, against which rests a string of three single-stranded cytosines (C67–C69) that form a continuous stack. In the strand opposite the cytosines, A66 makes a single sugar edge-Hoogsteen hydrogen bond to the second cytosine (C68) in the stack, completing a robust apical structure. Unexpectedly, although single-stranded, the C67–C69 stack assumes essentially the same helical structure as in an A-form duplex (Fig. 2a). This unique structure thus explains this region's perplexing sensitivity to both ssRNA-specific and dsRNA-specific RNases[25,29]. The unusual structural arrangement of ssRNA in the helical path of dsRNA observed in the crystal structure rationalizes the apparent dual nuclease sensitivity without necessarily invoking the presence of two highly stable yet interconverting conformers (Fig. 2b).

The sequence of the apical loop is largely constrained by the embedded Box B motif required for TFIIIC binding and the recruitment of RNA polymerase III to achieve its extraordinary abundance. Substituting C67–C70 with four uridines had little effect on VA-I activity (Fig. 2b, c; see the section "Methods" and Supplementary Fig. 5)[29]. Replacement of the apical loop by a flexible 10-nt U1A-binding loop slightly reduced its PKR-inhibitory activity, whereas a highly stable GAAA tetraloop significantly enhanced its potency (Fig. 2d, e). These results suggest that the duplex stability of the apical stem closing the loop, but not the loop sequence, is critical for PKR inhibition. This importance may be due to the binding of one of the PKR dsRBMs near the loop–stem junction, a model supported by RNA cleavage patterns by dsRBM1-tethered EDTA·Fe nucleases[37].

### A wobble-enriched apical stem of critical length.
The apical stem forms an underwound, imperfect A-form duplex of 22 base pairs (bp) that contains four interspersed wobble pairs (three G·U and one A$^+$·C; Fig. 1b and Supplementary Figs. 6 and 7). The length of the apical stem is critical for PKR inhibition. Extending the apical stem by 3 bp slightly enhanced VA-I activity, whereas mismatches or a side bulge (C75G::C76G[29]) strongly compromised it (Fig. 2c; Supplementary Fig. 8a, b). Step-wise shortening of the apical stem by 3, 6, and 10 bp led to progressive loss of inhibitory activity (Fig. 2d, e). The shortest construct (TetraΔ10bp, Fig. 2d) contained 12 bp of dsRNA in the apical stem, or 16 bp including the tetrastem extension, sufficient for binding a single dsRBM motif, and yet exhibited no activity against PKR. The requirement of at least 20-bp dsRNA (including the

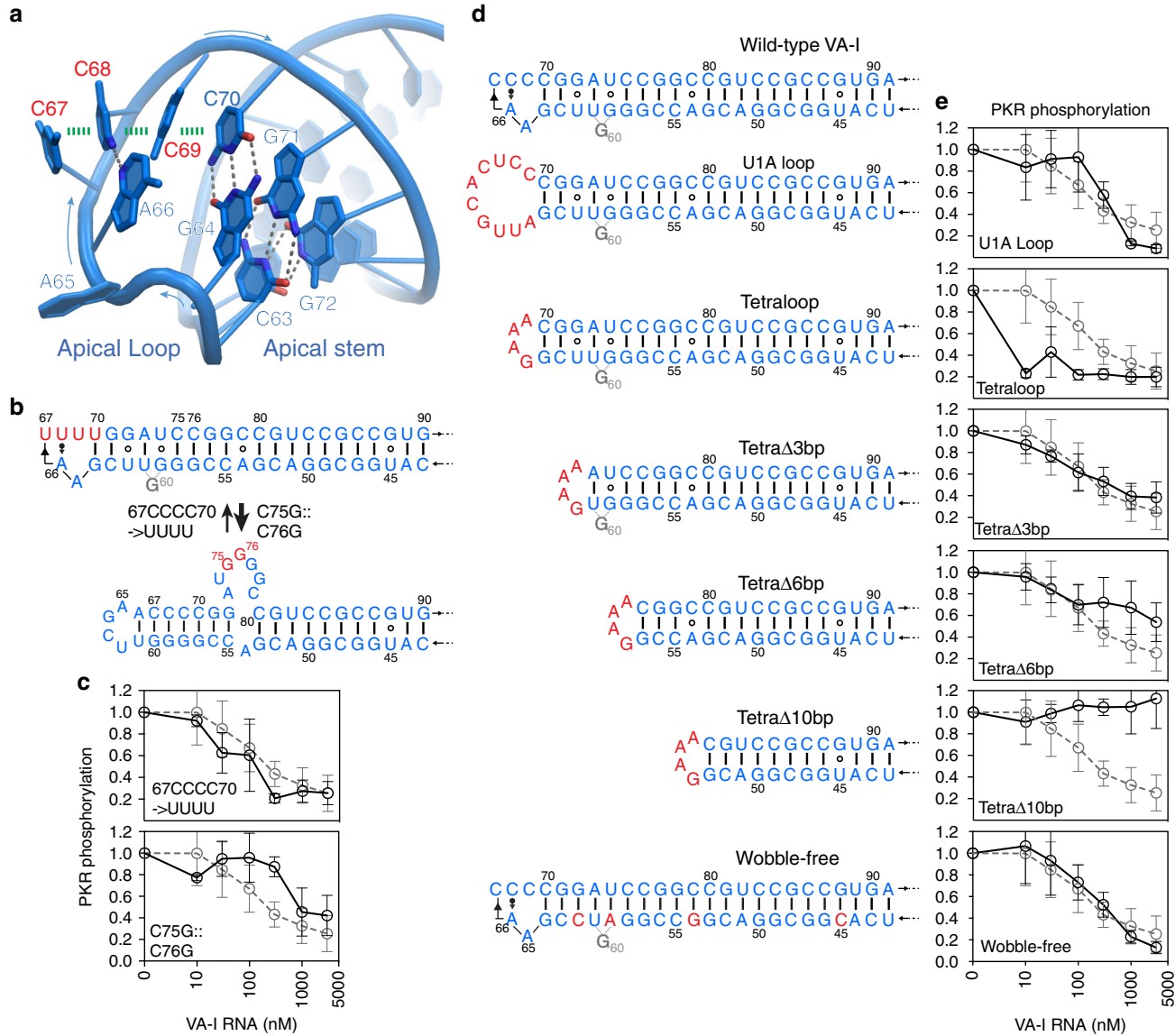

**Fig. 2** Structure and mutational analysis of VA-I apical region. **a** Apical region structure. The single-stranded, stacked C67–C69 in a dsRNA-like helical arrangement is indicated with red labels. Hydrogen bonds are denoted by gray dashed lines and base-stacking interactions are shown as green dashes. **b** Proposed alternative conformations of the apical region. The 67CCCC70 → UUUU substitution stabilizes the canonical conformation (upper panel) while a C75G::C76G substitution stabilizes the alternative, bulged conformation (lower panel). **c** Effects of the substitutions in **b** on PKR inhibition. The WT VA-I titration curve is shown as gray dashed lines for comparison in these and subsequent panels. **d** Secondary structures of wild-type and mutant VA-I RNAs harboring alterations to the apical loop, apical stem length, or the removal of apical stem wobble base pairs. The central domain is omitted for clarity. **e** Effects of VA-I alterations in **d** on PKR inhibition. Error bars represent s.d. from three independent replicates

tetrastem extension, e.g. TetraΔ6bp) to have partial activity suggests that both dsRBMs of a PKR monomer need to associate with the VA-I apical stem to achieve PKR inhibition, as suggested previously[38].

What is the role of the wobble base pairs that intersperse the apical stem and terminal stem? These conserved wobble pairs are also found in other adenoviral serotypes[3] and in VA-II (Supplementary Fig. 9). Wobble pairs alter the helical geometry and chemical composition of dsRNA grooves and can provide protein-binding determinants or anti-determinants[39]. To test the involvement of the wobble pairs in PKR inhibition, we constructed a wobble-free VA-I variant and found that it exhibited full inhibitory activity against PKR (Fig. 2d, e). Thus, the four wobble pairs in the apical stem do not contribute significantly to PKR inhibition. Instead, the wobble pairs seem to act to reduce the activation potential of the VA RNAs. Earlier

analyses found that nucleotide modifications to the minor groove (e.g., 2'-deoxy) or the Watson–Crick edge (e.g., 2-thio-U, 4-thio-U) strongly reduced PKR activation without significant effects on PKR binding, revealing an interesting nonequivalence of binding and activation[40]. Further, an insertion of 10 G·U wobble pairs in a 47-bp model dsRNA led to a major loss in PKR activation[40]. The VA-I terminal stem contains three additional G·U and another potential A+·C wobble pairs (Supplementary Fig. 1). In a previous study, the removal of VA-I apical stem or terminal stem wobbles led to enhanced PKR inhibition, and up to 10-fold increase in OAS1 activation[41], suggesting a key function of the wobbles in escaping OAS1 surveillance. Together, these findings suggest that viral RNAs can utilize genetically encoded wobble base pairs to produce chemical and geometric deviations from ideal dsRNAs to escape activation of dsRNA sensors such as OAS1 and PKR. This strategy is analogous to how post-transcriptional modifications of

host tRNAs and the inosine–uridine mismatches introduced by ADAR1 editing of dsRNA help avoid PKR activation[18,42], and has the added advantage of not requiring the cooperation of host modification enzymes.

**A core effector necessary and sufficient for PKR inhibition**. At the proximal end of the apical stem, A41 and U92 do not form ssRNA linkers as previously proposed[4]. Instead, they base pair to juxtapose the apical stem with the tetrastem to form an extended coaxial stack spanning 30 layers, just below the threshold of significant PKR activation (~33–35 bp; Figs. 1b and 3a)[12]. Thus, this region appears to function to capture PKR dsRBMs with affinity without leading to activation. The tetrastem, of the sequence GGGU/ACCC, is the most conserved region of VA-I, is also present in VA-II (Supplementary Fig. 9), and is a probable binding site of PKR based on footprinting analyses[25]. Mismatches in the tetrastem completely abrogate VA-I function (Fig. 3b–d), while compensatory mutations rescue only ~50% activity, confirming that the specific sequence is required in addition to secondary structure to elicit full inhibition[31]. Unexpectedly, our

structure revealed a base triple between the tetrastem (the G39–C120 pair) and C93 (Fig. 3c). C93 not only stabilizes the tetrastem, but also connects it to Stem 7 via two backbone hydrogen bonds to C94 and C120 (Fig. 3a–c). The bridging function of C93 is consistent with the significant impact of its substitutions on PKR-inhibitory activity (Fig. 3d). In silico mutagenesis showed that while C93U maintained the same base triple-forming 2-oxo group and exhibited essentially no defect, C93G was able to form an alternative base triple with G39 using its 2-amino group, actually enhancing its activity (Fig. 3d, Supplementary Fig. 10). In contrast, C93A lacked both functional groups at the C2 position, was presumably unable to form the triple, and exhibited significant defect (Fig. 3d). Conversely, in silico substitution of the G39–C120 pair with a C–G, A–U or U–A pair did not support formation of the base triple with C93, thereby driving in part the sequence conservation of the tetrastem (Supplementary Fig. 11). Additional drivers of the tetrastem sequence conservation may stem from a potential sequence-specific interaction between PKR's dsRBM2 and the tetrastem. As the dsRBM1 primarily binds to the apical region[37] and that the apical stem associates with both dsRBMs of PKR (see preceding

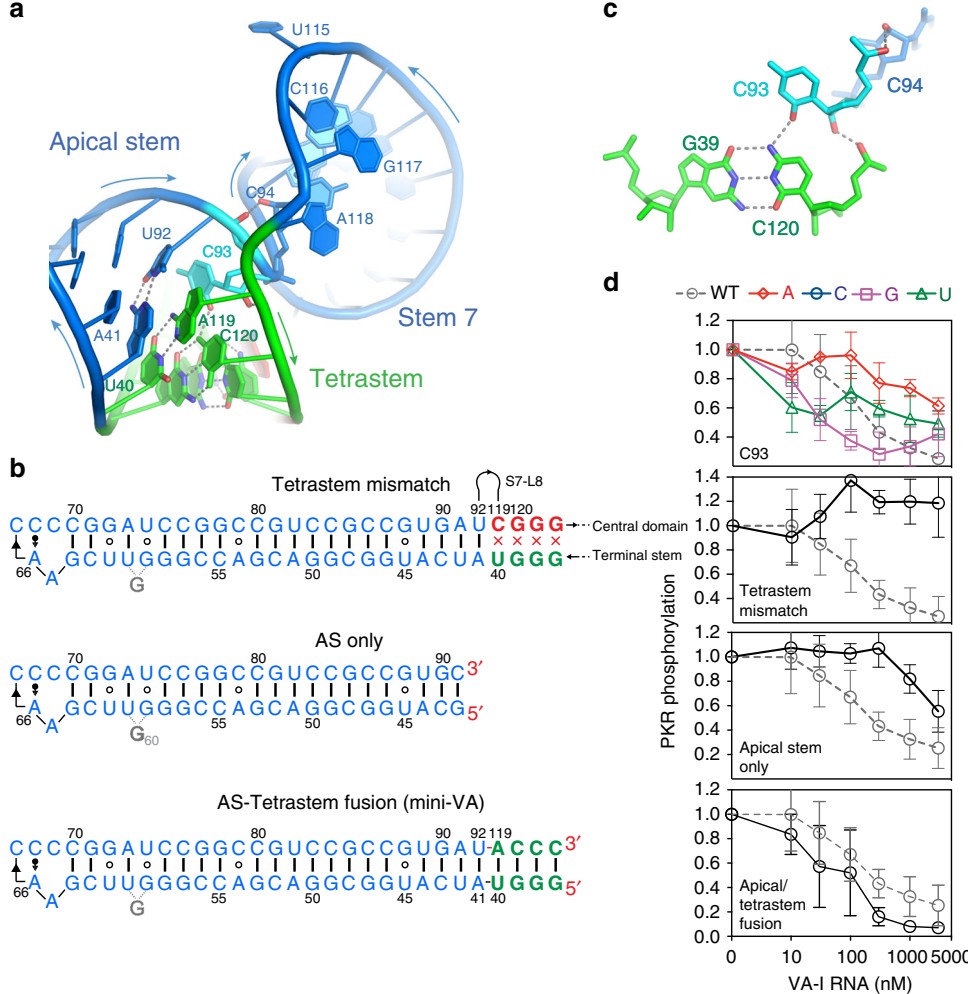

**Fig. 3** Stacked apical and tetra-stems from the PKR-inhibitory core. **a** Structure of the three-helix junction involving the coaxially stacked apical stem and tetrastem, and Stem 7. The junctional C93 is shown in cyan. **b** Secondary structures of a mutant VA-I harboring a mismatched tetrastem, a truncated apical stem (AS)-only construct, and a topologically engineered mini-VA RNA where the apical stem is fused to the tetrastem. The RNA termini are highlighted in red. **c** Structure of the G39–C120·C93 base triple. Note the two additional backbone hydrogen bonds from C93 to C120 and C94 connecting the tetrastem to Stem 7. **d** Effects of C93 substitutions and mutants in **c** on PKR inhibition. Individual titration curves for singly substituted mutants are colored based on the identity of the new nucleobase (A: red diamonds; C: blue circles; G: purple squares; U: green triangles). Error bars represent s.d. from three independent replicates

section), it is reasonable to speculate that the sequence of the tetrastem may serve to capture dsRBM2 which presumably binds to the proximal end of the stack.

Consistent with the functional importance of the tetrastem, the apical stem loop alone (26 stacked layers) exhibited minimal activity towards PKR (Fig. 3b–d). Remarkably, when we artificially fused the apical stem with the tetrastem by directly linking U92 to A119, the resulting fusion RNA (30 layers) exhibited full inhibitory activity (Fig. 3b–d). We define this topologically reengineered construct as the core effector for PKR inhibition and name it mini-VA. The 60-nt mini-VA is less than half of the length of VA RNAs, lacks the entire central domain, and yet exerts the full inhibitory effect. Why, then, is the central domain needed in VA-I? The central domain may provide a functional hub to topologically enable the coaxial stacking of the apical stem and tetrastem to present the core effector to target PKR, while simultaneously permitting the terminal stem to target Dicer, Exportin 5, or other host proteins.

**A pseudoknot-anchored central domain**. The proximal end of the tetrastem is affixed to the central domain through reciprocal nucleobase-2'-OH hydrogen bonds between A123 and G106, each capping a helix through cross-stand stacking (Fig. 4a, b). This cyclic dinucleotide linkage couples the core effector with the central domain. The solvent-accessible location of the unpaired, stacked A123 rationalizes its substantial SHAPE and $Tb^{3+}$ reactivity[30]. Interestingly, A123 employs two hydrogen bonds from its N1 and N3 to G106's 2'-OH and 2-exocyclic amine, respectively (Fig. 4b). These bonds likely underlie the pH sensitivity observed in thermal denaturation analyses, as the low-pH-specific cooperative transition corresponding to central domain unfolding was eliminated by an A123U substitution[30]. Consistent with the structural role of A123 in stabilizing the tetrastem and the extensive use of its nucleobase functional groups, its substitution with any other nucleoside led to significant drop in PKR-inhibitory activity (Fig. 4c). In contrast, substitutions of G106, on the central domain side, tend to enhance PKR inhibition (Fig. 4c). This observation hints at the functional specialization of the apical/tetrastems and the central domain.

Next we examined potential metal-binding sites on VA-I. Specific $Mg^{2+}$-binding sites were proposed based on strong sensitivities to $Tb^{3+}$-enhanced in-line cleavage near G97, A103, and A123 in the central domain[30]. Our structure does not detect strong structural $Mg^{2+}$ sites at these locations, consistent with essentially superimposable solution scattering curves in the presence and absence of $Mg^{2+}$ ions[33]. These observations suggest that specifically bound $Mg^{2+}$ ions, if any, are not required for VA-I structure formation. Nonetheless, our structure rationalizes the observed sensitivities to $Tb^{3+}$ based on their structural contexts. Like aforementioned A123, A103 is solvent exposed (Fig. 4d, e). Both adenosines likely attracted $Tb^{3+}$ due to their available N7 groups that are preferred $Tb^{3+}$-binding sites. Based on very prominent $Tb^{3+}$ cleavages at G97 and A123 and central domain structural sensitivity to solution pH, it was proposed that G97 may use its Hoogsteen edge to form a $A^{+} \cdot G$ long-range pair with a protonated A123[30]. Our structure showed that G97 is located far from A123 and actually base-paired to C111 in the middle of Stem 7 (Fig. 1b). We propose that the exceptional $Tb^{3+}$ sensitivity of G97 and its opposing strand is primarily attributable to the unique geometric and electrostatic properties of its neighboring U96·G112 wobble base pair (Fig. 1b; Supplementary Fig. 7f). G·U or U·G wobble pairs drive a ~2 Å outward shift of the uridine base into the major groove, forming a contiguous concave of electronegativity favorable for cation binding. In addition, the shifted U96 nucleobase is well situated to form a

cation–π interaction to stabilize a $Tb^{3+}$ bound to the N7 edge of G97 (Supplementary Fig. 12). Thirdly, the backbone geometry of wobble pairs exhibits higher plasticity and thus has a higher probability to engage the in-line configuration required for backbone scission. Similarly, the U45·G88 wobble pair region in the apical stem also exhibited strong cleavage both in the presence or absence of $Tb^{3+}$ ions[30]. These findings illustrate a potentially general utility in using $Tb^{3+}$ sensitivity to identify wobble base pairs in addition to locating specific $Mg^{2+}$ sites in RNA[43]. This is further consistent with preferred binding of $Ir^{3+}(NH_2)_6$ to tandem G·U pairs that has been used as a general strategy to obtain phase information for RNA crystals[44].

The central domain is anchored by a 3-bp pseudoknot formed between Loop 8 and Loop 10, as previously predicted based on sequence covariation, resistance to ssRNA-specific RNases, and reciprocal mutational analyses (Fig. 4d, e)[25,30,33]. Our structure further revealed that the pseudoknot is reinforced by stacking with G106 on top and extensive A-minor interactions with A36 (Fig. 4d–f). Reciprocally, the pseudoknot positions A36 along its minor groove and may dictate the orientation of the terminal stem that A36 caps, which in turn controls the overall shape or flexibility of VA-I RNA. As the global RNA shape is important for PKR control[34,35], the effect on overall shape may underlie the wide-ranging effects of A36 substitutions (Fig. 4g). A36U drastically compromised VA-I inhibition, presumably through loss of the global shape control. Interestingly, both the A36C and the previously reported G127U substitution extended the terminal stem by 1 bp through the creation of a 36–127 base pair, and both significantly enhanced VA-I activity against PKR. A reorganized junction structure between the terminal stem and central domain is likely responsible for the augmented activity.

The exact function of the central domain, especially of the long-presumed pseudoknot, has remained elusive. Depending on the exact nature of the alterations, pseudoknot disruptions exhibited variable, non-equivalent effects on PKR binding and inhibition[30,32,33]. Our targeted alteration of Loop 8 and Loop 10 produced minor defects in PKR inhibition, which were subsequently rescued by compensatory substitutions that restored base pairing (Fig. 4h). In comparison, alterations to Stem 7, adjacent to the pseudoknot, had drastic impact on VA-I activity (Fig. 4i)[28,32]. These findings suggest that the pseudoknot per se is not a primary functional element for PKR inhibition, but rather, plays ancillary roles. Based on our structure, the pseudoknot likely provides an anchor to position Stem 7, and topologically facilitates the coaxial stacking of the apical stem with the tetrastem, the key drivers of PKR inhibition. Thus, a direct fusion of the two stems in the mini-VA obviated the requirement of the central domain in PKR inhibition. Next we mapped the footprint of PKR onto the VA-I structure, which formed two contiguous areas on the same face with one cluster in the central domain (Fig. 5a, b). It is presently unclear if the putative PKR contact to the central domain is functionally important.

Unexpectedly, the Stem 7-Loop 8 structure of the central domain resembles the ASL of tRNAs (RMSD ~2.2 Å; Fig. 6a, b). Furthermore, the manner in which it pairs with Loop 10 and is stabilized by G106 stacking and extensive A-minor interactions is highly similar to the codon–anticodon interactions in the ribosome[45,46] and the specifier–anticodon interactions in the T-box riboswitches (Fig. 6c–e; Supplementary Fig. 13)[47,48]. Interestingly, while the ribosome and the T-box employ 3 and 2 RNAs, respectively, to form this structure, VA-I uses a single piece of RNA to achieve this geometry. The three independent occurrences of essentially the same structural configuration likely reflect a product of convergent evolution to construct a stable helix of minimal length. VA RNAs have been hypothesized to have originated from viral hijacking and concatenation of

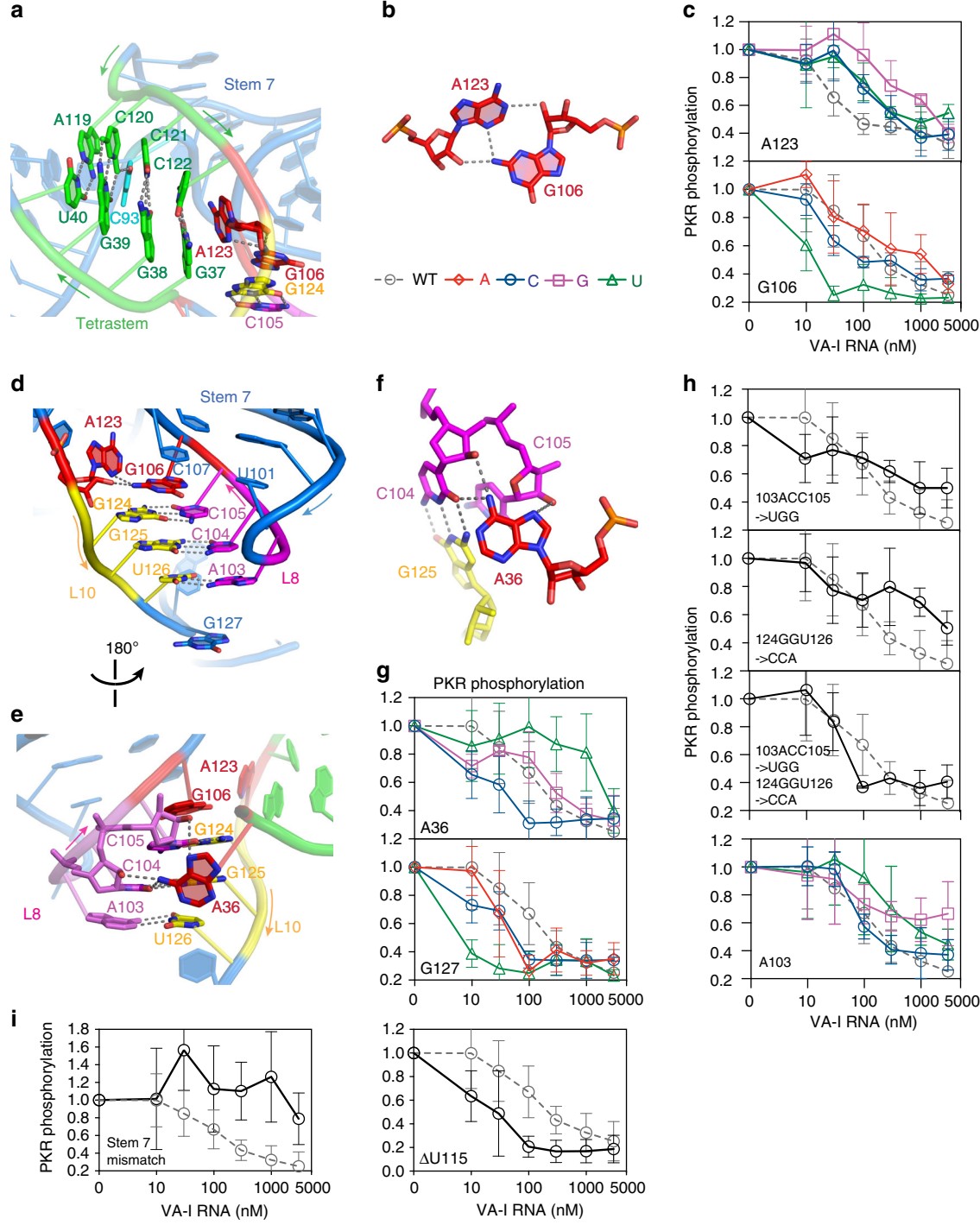

**Fig. 4** Structure of the central domain. **a** A123 stacks with and caps the proximal end of the tetrastem. **b** Interactions between A123 and G106 connects the tetrastem to the central domain. **c** Effects of A123 and G106 substitutions on PKR inhibition. **d–f** Structure of the Loop 8–Loop 10 pseudoknot and interactions with A36. **g** Effects of A36 and G127 substitutions on PKR inhibition. **h** Effects of pseudoknot disruptions on PKR inhibition. **i** Effects of Stem 7 alterations on PKR inhibition. Three mismatches in Stem 7 (G97U::G99U::G110U) abrogated PKR inhibition while ΔU115 enhanced VA-I potency. Error bars represent s.d. from three independent replicates

neighboring tRNAs genes, based on the presence of Box A and B motifs[49]. We further found that the nucleoside identities of Stem 7-Loop 8 satisfy the criteria delineated in a compendium of 382 elongator tRNAs and contain an ACC anticodon for glycine, one of the primordial amino acids (Fig. 6a)[50]. The clear sequence and structural similarities support the tRNA origin hypothesis for VA RNAs and may also hint at a functional role of the apparent tRNA mimicry. Similar ASL-like structures are used by plant viruses to enhance translation or facilitate replication[51], and by HIV-I to recruit human lysyl-tRNA synthetases to facilitate replication initiation[52].

The structural constraints derived from the VA-I crystal structure allowed more accurate structural alignment of VA-I and VA-II using Dynalign II (Supplementary Fig. 9)[53]. VA-II can assume an overall secondary structure very similar to VA-I, albeit with major interruptions to its apical stem. VA-II also

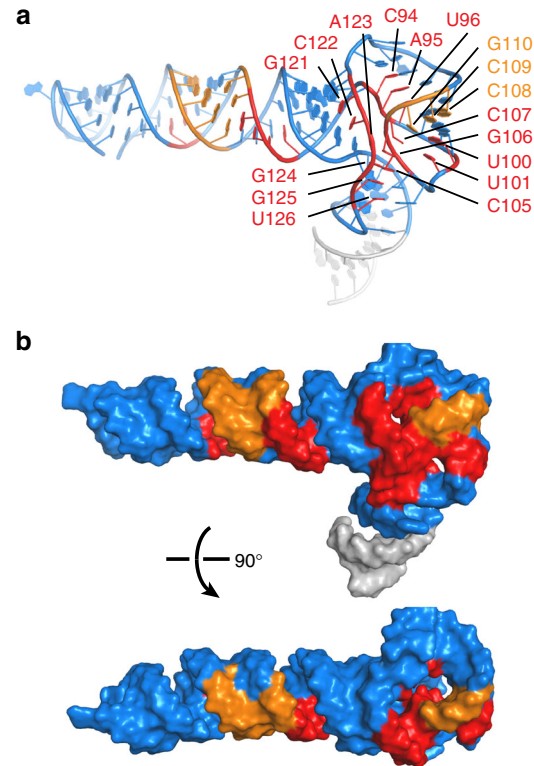

**Fig. 5** PKR footprint on VA-I. **a** Cartoon illustration of VA-I residues protected by PKR binding from RNase digestion (red) and chemical probes (orange)[61]. Nucleotides 86–89 are protected from both. **b** Surface representations of (**a**) show two contiguous binding surfaces located in the apical stem and central domain on the same face of VA-I

presumably lacks the central domain pseudoknot. Its tetrastem appears to be 5 bp in length with essentially the same sequence as in VA-I. The interruptions in the apical stem resemble the AS bulge insertion variants that reduced PKR inhibition (Supplementary Figs. 8 and 9) and are probably responsible for VA-II's minimal PKR-inhibitory activity observed in vivo[54]. Instead of targeting PKR, VA-II which originated from a gene-duplication event from VA-I, is better suited to target other host enzymes, such as Dicer[5] and ADAR.

## Discussion

The sharply bent, coaxially stacked, wobble-enriched, and pseudoknot-anchored VA-I structure provides a first glimpse into how viruses use extremely abundant, structurally compact RNA devices to compromise the host immune system to allow viral propagation. The structure reveals a collection of salient global and local features that are congruent with VA-I's role as an exemplary PKR inhibitor made of RNA, and provides a lens through which to understand the multifaceted allosteric mechanisms of PKR activation and suppression[24,38]. Below, we discuss the contributions of its overall shape, wobble pairs, and central domain pseudoknot to VA-I function, as well as its utility in biotechnological applications.

Our crystal structure captured VA-I in an acutely bent "V" shape (~60°), whereas SAXS envelopes generally showed obtuse angles between the apical and terminal stems[32,33]. There are at least two mutually non-exclusive possibilities for this difference. First, crystal packing could have altered the structure locally or globally. To understand the potential influence of the crystal-packing contacts, we analyzed the packing interfaces (Supplementary Fig. 14). A primary packing contact occurs at the top of

Stem 7, where the single-stranded trinucleotide linker C116–G117–A118 pairs and stacks with its crystallographic symmetry mate. Interestingly, G117 bridges C116 and A118 of its symmetry mate forming a base triple and the two triples stack against each other, forming a presumably stable crystal-packing contact. As a previous SHAPE analysis confirmed the single-stranded nature of the trinucleotide linker, it seems unlikely that this packing contact drastically altered the overall structure. Nonetheless, the local structure of the trinucleotide linker that caps Stem 7 may have been locally deformed. A second packing contact occurs between the terminal G26 residue and its symmetry mate. This single, non-specific stacking interaction does not seem capable of drastically impacting the RNA conformation. Lastly, G127 intercalates between G64 and A66 of the apical loop. In this region, SHAPE analyses and nuclease sensitivities are consistent with the crystal structure. Further, given the numerous specific contacts within the central domain, it is likely that a drastic pivoting motion of the terminal stem would disrupt the pseudoknot, whose formation was well supported by multiple methods. Alternatively, the differences could have stemmed from the limited resolution of the solution technique, inaccuracies within the proposed secondary structure models, a scarcity of distinguishing features of the SAXS envelops that could sufficiently constrain modeling, as well as an incomplete understanding of the hydration and scattering properties of nucleic acids (especially at the higher angles of scattering) compared to those of proteins for which the ab initio reconstruction algorithms were parameterized[55]. Curiously, similar to the differences between the SAXS envelopes and X-ray structure of VA-I, the SAXS envelops of tRNAs and tRNA-like viral RNA structures consistently showed obtuse, ~120° angles between the two arms of the L shape, as opposed to the canonical ~90° angles observed by X-ray crystallography and cryo-EM[56]. In solution and without the constraints of the crystal-packing lattice, the VA-I RNA is likely more flexible and can sample additional conformations, such as the more obtuse angles, or even transiently disengage its pseudoknot interaction. It is presently unknown if such structural flexibility is important for VA-I's ability to target more than one host proteins.

One of the prominent local features of the VA-I structure is the over-representation of wobble base pairs that populate the apical stem, terminal stem, and even Stem 7 (Fig. 1b; Supplementary Fig. 1). Interestingly, both the Epstein-Barr virus-encoded small RNA 1 (EBER1) Stem IV, the primary binding site of PKR dsRBDs[57,58], and the nc886 RNA[22], a recently described human noncoding RNA that inhibits PKR, are similarly enriched for wobble pairs. Thus, the convergent use of wobble pairs in at least three PKR-inhibitory RNAs may reflect a general strategy to construct a pseudo-activator—an inhibitory device that is disguised as an activator. dsRNAs interspersed with wobble pairs, as well as other helical imperfections, may sufficiently mimic ideal dsRNAs to recruit and trap their target enzymes such as PKR and OAS1, but employ subtle structural or chemical deviations to misalign multimerization interfaces or to distort catalytic geometries, ultimately leading to non-activation or inhibition.

It was shown that the wobble base pairs allowed VA RNAs to escape OAS1 and PKR activation at the expense of inhibitory potency against PKR[41]. Given the extreme abundance of VA RNAs, there is little incentive to create the strongest possible PKR inhibitor. Thus, priority was given to guard against inadvertent activation by misfolded or degraded viral RNAs, which can lead to catastrophic restriction of the viruses. It is clear that the wild-type VA-I is not optimized for maximized PKR-inhibitory activity, as a number of substitutions that we and others tested actually enhanced PKR inhibition. In addition to their effects on PKR and OAS1, the wobble pairs may also impact other targets of

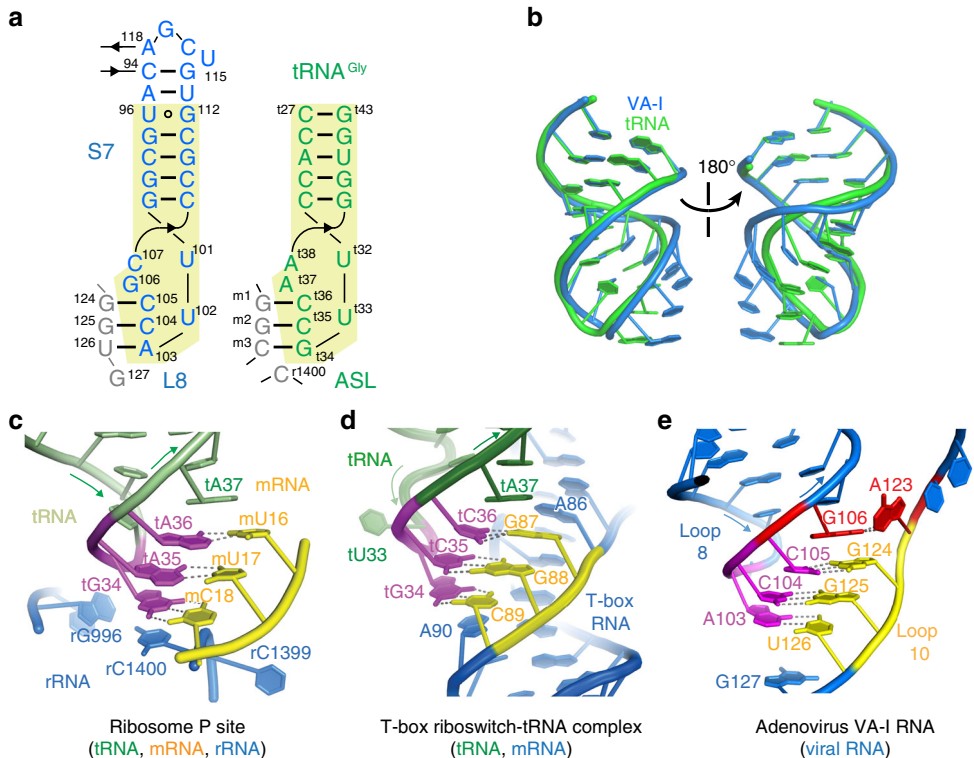

**Fig. 6** Structural similarities with the tRNA ASL and codon–anticodon interactions. **a** Secondary structures of VA-I S7-L8 region and a tRNA^Gly ASL. Their base-pairing partners, VA-I L10, mRNA codon, and rRNA are in gray. Numbering of tRNA, mRNA, and rRNA are preceded by "t", "m", and "r", respectively. **b** Overlay of the tertiary structures. Overall RMSD is 2.2 Å. **c** Structural configuration of the codon–anticodon interactions in the P-site of the ribosome (PDB: 4V5D[https://www.rcsb.org/structure/4v5d])[45,46]. The tRNA, mRNA, rRNA are shown in green, yellow, and blue, respectively. **d** A nearly identical structural configuration was observed in T-box riboswitch-tRNA co-crystal structures (PDB: 4LCK[https://www.rcsb.org/structure/4lck])[47,48]. Note the cross-strand stacking by conserved purines (tRNA t37 and T-box A90). **e** Codon–anticodon-like interactions in the VA-I RNA central domain. The 3-bp duplex is presumably stabilized by stacking by G106, which occupies a position equivalent to tRNA purine t37

VA RNAs. The A+54·C79 wobble and another potential A +5·C151 wobble pair in the terminal stem likely constitute preferred ADAR-editing sites and may contribute to its capture or inhibition. Therefore, the evolution of VA sequence and structure was likely constrained by selective pressures from multiple host targets, each with distinct RNA-binding specificities. The simultaneous selection of convergent and divergent features produced an extant multifunctional RNA that contains an amalgam of balanced features and characteristics[4].

In contrast to its auxiliary role in PKR inhibition through facilitating RNA folding, the central domain pseudoknot is crucial for preventing PKR activation by a full-length VA-I. This is evidenced by the observation that pseudoknot disruptions consistently converted VA-I into robust PKR activators[32], an outcome counter to VA-I's intended function. There are at least three possible, mutually non-exclusive mechanisms by which the central domain prevents PKR activation. First, the central domain pseudoknot may function to constrain the overall shape or flexibility of VA-I to block PKR activation. The multiple targets and functions of VA-I likely required a longer structural platform than its PKR-inhibitory core (mini-VA) can provide. The extant longer VA-I scaffold was proposed to have originated from concatenated host tRNA genes and our observed structural similarities to tRNAs support this notion[3]. Such longer RNA platforms, especially those that are near-linear in architecture or contain flexible regions, carry substantial risks of activating PKR or other dsRNA sensors. This is exacerbated by PKR's ability to straighten bent, bulged, and other non-contiguous dsRNA helical segments and stack them into extended platforms that generally activate PKR[34,59]. For host RNAs, such risks are mitigated by

shielding by RNA-binding proteins or editing by ADAR, etc.[18]. For VA-I, the pseudoknot appears to limit its conformational freedom and generate an acute bend in the dsRNA trajectory, thus antagonizing PKR's structure-remodeling activity to avoid induced activation. This notion is consistent with the lowest temperature factors of this region (Supplementary Fig. 4a–d). When the pseudoknot was unable to form, the resulting malleable RNA structure activated PKR. Second, the central domain pseudoknot may function as a steric barrier against deposition of a second PKR monomer, thus blocking PKR dimerization and activation on the same VA-I RNA. This explanation is supported by previous biophysical analyses of PKR–VA-I complexes using analytical ultracentrifugation[33] and dynamic light scattering[32,38]. We further confirmed that PKR binds 1:1 to full-length and ΔTS VA-I RNAs using SEC–MALS analyses (Supplementary Fig. 15). Finally, the VA-I central domain may suppress PKR activation through direct contacts to the kinase domain or its adjacent basic patch in the inter-domain linker[60]. This notion is supported by the protection of the central domain from nucleases and chemical agents by PKR (Fig. 5)[61], and 4-thio-U crosslinking between an ssRNA–dsRNA activator and the basic patch[60]. The precise mechanisms by which the VA-I central domain blocks PKR activation await structural and biochemical analyses of PKR–VA-I complexes. It is further possible that the VA-I central domain uses its unique 3D structure to execute pro-viral functions that have not been well-characterized, such as the inhibition of RNA helicases.

The structural elucidation of VA-I RNA informs the development and optimization of VA RNAs as valuable tools in biotechnology applications, adenovirus-based gene therapies, and

vaccine development, etc. Co-transfection of VA genes in mammalian cells (e.g. the Promega pAdVantage vector) overcomes PKR-mediated translation inhibition and dramatically increases protein expression by 10–70 fold[62]. The 60-nt mini-VA RNA not only exhibits full inhibitory activity against PKR to boost protein translation, but also will permit simultaneous use of therapeutic small hairpin RNAs (shRNAs) for gene silencing, as it lacks the regions that target Dicer and Exportin-5. Conversely, if normal translation activity is desired, a truncated VA-I RNA with a shortened apical stem will likely only target the RNAi machinery while sparing PKR and translation. Ectopic expression of VA RNAs down-regulates PKR activity and can be employed to counteract excessive activation of the interferon response and to protect tissues from apoptosis and inflammation. VA-I also holds exceptional promise as a versatile vehicle to exogenously express and fold RNA fragments of interest similar to the widely used tRNA scaffold in which a cargo RNA is inserted in the ASL[63]. The extraordinary potency of VA transcription, proven cytoplasmic delivery and stability, and built-in suppression of dsRNA cargo-triggered immune response, make for an ideal RNA-expression platform.

As the quintessential viral noncoding RNA, the specific structural features of the multifunctional adenovirus VA-I provide a starting point and reference to understand a growing collection of highly structured endogenous (e.g., IFN-γ and TNF-α) and exogenous (e.g., EBERs) RNA modulators of host proteins, such as PKR, OAS1, and others. Comparative analyses of exemplary activating and inhibitory RNAs are expected to clarify the principles of self vs. non-self RNA discrimination[64,65] and inform the design of functional RNA devices that elicit specific desired cellular responses.

## Methods

**Preparation of RNA and crystals**. The adenovirus serotype 2 VA-I RNA was modified by removing the functionally dispensable terminal stem and loop 2 (nucleotides 1–31 and 132–161) and appending a 6-bp stem (sequences 5'-GGACCU and 5'-AGGUCC, Fig. 1a, b). The resulting ΔTS VA-I RNA (112 nts) was transcribed in vitro by T7 RNA polymerase, purified by electrophoresis on 10% polyacrylamide, 8 M urea gels (29:1 acrylamide:bisacrylamide), electroeluted, washed once with 1 M KCl, desalted by ultrafiltration, and stored at −20 °C[47]. For crystallization, 4.0–8.0 mg/mL (112–223 μM) VA-I RNA in RNA folding buffer consisting of 10 mM Tris–HCl (pH 7.5), 50 mM KCl, 5 mM MgCl$_2$, and 1 mM spermine was mixed 1:1 with a reservoir solution composed of 50 mM sodium cacodylate (pH 6.5) and 28–30% 2-methyl-2,4-pentanediol (MPD) in a hanging-drop vapor diffusion tray. Single, obelisk crystals grow in 1–2 days and reach maximum dimensions of ~500 × 50 × 30 μm³ after 18–21 days. RNA crystals were cryo-protected in a synthetic mother liquor that contains 40% MPD and vitrified in liquid nitrogen at 100 K. Crystals exhibited the symmetry of space group P4$_1$2$_1$2 and unit cell dimensions are shown in Table 1 and Supplementary Table 1. For de novo phasing, crystals were soaked in the cryoprotection solution supplemented with 5–20 mM Ir(III) hexammine for 2–16 h before vitrification.

**Data collection and structure determination**. Single-wavelength anomalous dispersion (SAD) data were collected near the Iridium L1 edge (0.9218 Å) at the SER-CAT beamline ID-22 at the advanced photon source (APS). Initial SAD phasing using single datasets did not correctly identify the Ir substructure. A total of 16 Ir-SAD datasets with diffractions extending to 3–5 Å were analyzed by hierarchical agglomerative program BLEND[66] (Supplementary Fig. 2 and Supplementary Table 1). One of the clusters consisting of nine datasets exhibited strong isomorphism, as evidenced by a linear cell variation (LCV) value of 1.14% (or 1.9 Å). The cluster of nine datasets were merged by BLEND in synthesis mode into a single dataset at 4.0 Å, which exhibited drastically improved half-data-set anomalous correlation (Ano CC$_{1/2}$) from 0.2 to 0.8 in the highest resolution shells. Ir substructure was readily identified by SHELX[67] and Phenix.HySS[68], and refined by Phenix.Autosol, producing a figure of merit (FOM) of 0.56. RNA helices and nucleosides were clearly visible in the resulting experimental electron density map. Model building was performed in Coot[69], as guided by iteratively generated MR-SAD maps using an evolving model. Finally, the near complete model was located in the native dataset using PHASER[70], manually adjusted and rebuilt using Coot, and refined using Phenix.Refine. The model was then corrected by ERRASER[71] and further refined (Table 1).

**Protein expression and purification**. N-terminally his-tagged, dephosphorylated human PKR was heterologously expressed and purified from Rosetta™ 2(DE3) pLysS cells co-transformed with pET28a-HisPKR[72] and pPET-PKR/PPase[73] plasmids. The latter plasmid was used to provide a source of lambda phosphatase while the expressed untagged PKR derived from this plasmid was removed during the first Ni²⁺ column step. Cells were initially cultured in ampicillin-supplemented and kanamycin-supplemented terrific broth at 37 °C until OD$_{600}$ reached 1.0. Then the cells were shifted to 30 °C and 0.5 mM IPTG was added to induce protein expression for additional 1.5 h before harvesting. Cells were resuspended in a lysis buffer comprised of 20 mM HEPES–KCl (pH 7.5), 200 mM NaCl, 0.1 mM EDTA, 5% glycerol, 5 mM β-mercaptoethanol, 10 mM Imidazole, supplemented with SIGMAFAST™ Protease Inhibitor Cocktail, lysed using a microfluidizer, clarified by centrifugation, and the supernatant loaded onto a HisTrap HP Ni²⁺ column on an ÄKTA pure chromatography system. Fractions containing PKR were pooled and further purified on a HiTrap Heparin HP column to remove associated nucleic acids, and was dialyzed into a storage buffer consisting of 20 mM Tris–HCl (pH 7.5), 100 mM KCl, 2 mM MgCl$_2$, 20% glycerol, and 8 mM β-mercaptoethanol and stored at −80 °C. The purity and mass of the purified PKR was verified by SDS–PAGE and liquid chromatography–electrospray ionization–quadrupole-time of flight–mass spectrometry (LC–ESI–Q-TOF–MS), yielding an intact protein mass of 66,240.5 Da, as compared to the calculated mass of 66,239.8 Da (Δmass = 0.7 Da). To further confirm that the purified PKR carries no removable phosphates, PKR was treated with lambda phosphatase (NEB), which led to no change in intact protein mass or phosphate content as assessed by the ratio of Pro-Q Diamond Phosphoprotein Gel Stain (Thermo-Fisher) signal and SYPRO Ruby Protein Gel Stain signal of the same gel.

**PKR kinase activity assay**. To measure VA-I's effects on PKR kinase activity, 150 nM PKR was first incubated with wild-type or mutant VA-I RNAs ranging from 0 to 3000 nM in a buffer containing 20 mM HEPES–NaOH (pH 7.5), 50 mM NaCl, 4 mM MgCl$_2$, 5% glycerol, and 2 mM DTT for 10 min at room temperature with gentle agitation. A 5× mixture of 250 μM ATP and 0.15 μg/mL poly I:C was then added and incubated at 30 °C for 30 min to challenge the PKR–VA-I complex to induce PKR autophosphorylation. Reactions were quenched by addition of an equal volume of 2.0% SDS in Tris-buffered saline (TBS, pH 7.4), and subsequently applied to a 48-well Bio-Dot SF (Bio-Rad) microfiltration system containing 400 μL of 0.1% SDS in TBS in each well. PKR was adsorbed onto a nitrocellulose membrane under vacuum. PKR phosphorylation was assessed by western blot analysis using Anti-P-PKR (phospho T446) antibody (rabbit, ABcam ab32036, 1:3000 dilution) and Alexa Fluor 488-conjugated anti-rabbit secondary antibody (goat, Invitrogen, 1:15,000 dilution). Additionally, total PKR was quantified on a separate slot blot using Anti-PKR antibody (mouse, Santa Cruz Biotechnology sc-6282, 1:1300 dilution) and Alexa Fluor 488 conjugated anti-mouse secondary antibody (goat, Invitrogen, with 1:5000 dilution). Band intensities were captured using a GE Typhoon Trio+ Imager and quantified using ImageJ. The ratios of anti-P-PKR (phospho T446) band intensities and anti-PKR intensities were used to assess PKR phosphorylation, and were normalized against the control reaction containing poly I:C but no VA-I RNA. At least triplicate experiments were performed for each RNA. In addition to the slot blot analyses, select reactions are also electrophoretically separated on SDS–PAGE, transferred to PVDF membranes using an iBlot 2 Dry Blotting System (Thermo-Fisher), and quantified by immunoblotting similar to the slot blots (Supplementary Fig. 5). Both assays yield comparable results. Substitutions in VA-I RNA were introduced by QuikChange Lightning Site-Directed Mutagenesis Kit (Agilent) and verified by Sanger sequencing of the plasmids. The presence of desired substitutions in the RNA was confirmed by reverse transcription, PCR amplification of the cDNA, followed by Sanger sequencing.

**Sequence and structural alignment of VA-I and VA-II**. RNA sequences of Ad2 VA-I and VA-II were aligned and folded together by Dynalign II[53] of the RNAstructure suite, using structural constraints derived from the crystal structure of the apical and central domains of VA-I, to produce an updated secondary structure model for VA-II and a consensus secondary structure of both VAs (Supplementary Fig. 9).

**Light-scattering analysis**. Size-exclusion chromatography with multi-angle static light scattering (SEC–MALS) was used to assess the binding stoichiometry of PKR to WT and ΔTS VA-I RNAs. 30 μM of individual RNA or protein component or their equimolar mixtures were incubated for 15 min in a buffer consisting of 25 mM Tris–HCl pH 7.5, 150 mM NaCl, and 2 mM MgCl$_2$ on ice prior to injection on a Superdex 200 Increase 10/300 GL column using an Agilent HPLC system. The HPLC system was coupled to a DAWN HELEOSII detector equipped with a quasi-elastic light scattering module and an Optilab T-rEX refractometer (Wyatt Technology). Data were analyzed using the ASTRA 7.1.4 software (Wyatt Technology Europe).

**Reporting summary**. Further information on research design is available in the Nature Research Reporting Summary linked to this article.

## Data availability

Atomic coordinates and structure factor amplitudes for the VA-I RNA have been deposited at the Protein Data Bank under accession code 6OL3 [https://www.rcsb.org/structure/6OL3]. All other data generated or analyzed during this study are included in this published article (and its supplementary information files) and available from the corresponding author upon reasonable request.

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

## Acknowledgements

We thank S. Li for sharing RNA samples, technical assistance, and select reagents, I. Botos for computational support, T.E. Dever for advice and sharing strains and plasmids, G. Piszczek and D. Wu for support in biophysical analyses, Y. He and N. Tjandra for fermentation support, Y.H. Xing and L.L. Chen, D.-Y. Lee, and R. Levine for help with mass spectrometry, W. Zhang and J. W. Szostak for a gift of Ir(III) Hexammine, J. Gerton and M. Mattingly for a gift of pET28a-HisPKR plasmid, and F. Gu, K.C. Suddala, A. Ferré-D'Amaré, A. Roll-Mecak, and M. Palangat for discussions. Data were collected at Southeast Regional Collaborative Access Team (SER-CAT) 22-ID beamline at the Advanced Photon Source of the Argonne National Laboratory, supported by the U.S. Department of Energy under Contract no. W-31-109-Eng-38. This work was supported by the intramural research program of NIDDK, NIH.

## Author contributions

J.Z. conceived the study and designed crystallization constructs. I.V.H., J.M.G., C.B.-N. and F.E.H., prepared various RNAs. I.V.H. prepared crystals and collected diffraction data. I.V.H. and J.Z. determined the structure. C.B.-N., S.B. and J.Z. purified PKR. J.M.G., F.E.H. and C.B.-N. conducted biochemical assays. C.B.-N. and J.Z. designed the mini-VA. J.Z., J.M.G., C.B.N. and F.E.H. wrote the paper.

## Additional information

**Competing interests:** The authors declare no competing interests.

