## [Peer Review File · Nature Communications]

Reviewers' comments:

Reviewer #1 (Remarks to the Author):

This manuscript reports a beautiful 2.7 Å crystal structure of the VA-I RNA. The structure of VA-I has long been sought after and will be well received by those in the field. The resulting structure shows an acutely bend RNA with several interesting features. The apical stem coaxially stacks with a four base pair stem near the central core. The central core features a compact pseudoknot structure that results in the acute bend between the apical stem and the truncated terminal stem. The manuscript examines the apical stem, tetrastem, and central domain structures and provides functional data to probe the importance of select interactions and features on PKR inhibition. The results suggest that the VA-I coaxial-stacked apical stem/tetrastem structure is responsible for PKR inhibition and the authors are able to design a fusion fragment that recapitulates wild-type inhibition. The manuscript represents a breakthrough in this field, and the structural data are of high quality. As such, the manuscript deserves publication in Nature Communications if the concerns below are addressed in a revised manuscript.

Major

1. The major shortcoming of the paper, which I believe can be easily remedied, is that the authors fail to present a cohesive story in relating the structural features of VA-I to their functions. The author's functional data mainly focuses on manipulating the apical stem length/sequence and shows that an apical/tetrastem fusion is capable of inhibiting PKR in the absence of the central domain. However they later offer hypotheses that suggest inhibitory functions for the central domain and the resulting bend. This contradicts their apical/tetrastem fusion RNA data that suggests a Wobble-rich stem, with no bend, is sufficient to inhibit PKR. They also suggest that the central domain may prevent PKR dimerization, however I do not believe that their functional data on the fusion RNA supports this hypothesis. The authors do say that their data supports a model in which the pseudoknot's main role is in stabilizing the coaxial stacking, but I feel it gets lost and minimized in the later statements. Simply cleaning up this region of text (Lines 307-320) and presenting a more cohesive story would greatly improve the clarity of the manuscript.

2. The error bars for the inhibition data are not defined (standard error or standard deviation?). My eyes may be deceiving me, but some of the error bars on the inhibition plots for the mutants appear quite large and overlap with the error bars of the wild-type control. Because of this, readers cannot fully trust some of the trends seen. It would be good if the authors could collect additional replicates in order to make their functional data more robust. A few of the mutants that appear to have issues:

- a. Tetra $\Delta 6$ bp
- b. Tetra $\Delta 3$ bp
- c. AS only

3. The authors give a single line suggesting crystal packing may have contributed to the differences between the solution SAXS and x-ray crystallography data. I believe that the authors should expand upon this and inform readers about this possibility and why it's so quickly dismissed as a possibility. The section on the "tRNA codon-anticodon mimicry" could be replaced with a new small section in the manuscript that describes the crystal packing and implications. Many RNA crystal structures have exhibited crystal packing interactions, including stabilizing "kissing loop" interactions. Several figures (at least supplemental) on this should also be included.

4. The authors should have a crisper discussion of how their structural data explains PKR inhibition. Previous work by Puglisi and Cole had showed that VAI binding blocks dimerization of PKR required for activation. The authors hint at this, but a clear section in the discussion is needed.

Minor

1. The authors needlessly expand upon the "codon-anticodon" similarity of the central domain.

This region of structure is indeed of interest, but this section feels more like an attempt to artificially increase the interest in the work. I would recommend that the “kissing” interaction should just be included within the section on the central domain rather than have its own section, and move the figure to the supplementary material

2. The writing, especially in the introduction, is hampered by the large amount of acronyms and the author’s tendency to list many examples. Making a statement and supporting it with citations would clean up the writing immensely and make it more accessible.

3. The writing also contains instances of superfluous wording. For example, phrases like “necessary and sufficient” (Line 105) can be simplified for clarity and directness. Just say that the RNA construct is derived from the Dicer-processed form and contains all the elements necessary for PKR inhibition.

4. Figure 2 a/b does not clearly illustrate (to me) how the current structure alleviates the old RNase data. A reader unfamiliar with the literature on how these conformers were stabilized will likely be confused. This may just require some clarity in writing and not a new figure.

5. Line 359: I believe the authors meant “(~60°)” and not “(~120°)”.

6. Line 419 about targeting a cavity in VA-I feels very forced and another attempt to artificially increase the impact of the paper. This is the very first instance that this structural feature is described (in the discussion!). This feature should either be expanded on (with implications, comparisons with other RNAs, etc) or the comment about it removed.

Reviewer #2 (Remarks to the Author):

The authors present a novel high-resolution (2.7Å) structure of the adenovirus Virus-Associated RNA VA-1. This model helps to explain previously ambiguous experimental data but it also has notable differences from previously reported models, particularly from a model generated through SAXS. The authors use this new data to explain and to further explore the functional determinants of the molecule’s inhibitory activity against a particular protein target, PKR. However it is also clear that the activity of VA-1 relative to PKR is not the only significant selection pressure to evolve the VA-1’s particular overall structure as they describe it (A more thorough investigation of these other pressures may be beyond the scope of the manuscript). The authors use structural homology and present limited sequence alignment data to hint at the evolutionary history of the molecule. The authors touch upon interesting synthetic biology applications that can be further considered as a result of this work. Given that there are relatively few high-resolution structures of viral subdomains and that this work yields insight into viral innate immune system evasion, the work is broadly important in my opinion.

General and specific comments follow:

About PKR binding to VA-I domain: The authors show the impact of mutations/deletions on PKR inhibition/activation. It would have been informative to have information on PKR binding as well.

About VA-I and VA-II: The authors discussed in introduction that there are two categories of these RNAs, VA-I and VA-II. What are the similarities and differences in these RNAs? Does the structure of VA-I tell anything about VA-II? A comparative analysis would be interesting.

About X-ray data and refinement: The Rmerge for the highest resolution cell is 364% and I/σ(I) is 0.82. It means the meaningful resolution might not be 2.7-Å. What is CC1/2 for the overall data and for the highest resolution shell?

Line 183: Please comment on the conservation of the wobble pairs?

Line 190: Shouldn’t the sentence read “nonequivalence of binding and activation”?

Line 241: Mention that there could be as yet unassigned functions to the central domain

Line 249: The authors state that "A123 employs a protonated N1 hydrogen bond to G106" but no protonation needs to be invoked. N3 can accept a hydrogen bond from the exocyclic amine of G106 and N1 can accept a hydrogen bond from G106 2'-hydroxyl group.

Lines 269-277: Regarding Tb³⁺ cleavage, in my opinion this assay is a form of in-line probing with Tb³⁺ acting as a catalyst to accelerate cleavage. In-line probing cleavage sites identify residues in the RNA that are flexible, so that the 2'-OH can get in line with the adjacent P-O5' bond. I disagree that it is simply a measure of solvent accessibility. Riboses in duplex regions are solvent accessible but are not good sites for Tb³⁺ cleavage for example.

Line 307: This section is confusing. Above, the authors state that the central domain can be removed and show that the AS-Tetrahem fusion has inhibition properties that exceed the wild-type. Yet, here the authors try to ascribe a role for the central domain inhibition versus activation of PKR. What is the reason for this discrepancy? Does the role of the central domain become important only in nonfused tetrahem constructs?

Discussion: Perhaps state that it is possible that the domain has some flexibility with respect to being acutely bent versus more obtuse. Flexibility is frequently a functional feature of structured RNAs.

Figure 2: It would be best to make all of the y-axes the same for each of the inhibition plots.

Point-by-point responses to Reviewers' comments:

Reviewer #1 (Remarks to the Author):

This manuscript reports a beautiful 2.7 Å crystal structure of the VA-I RNA. The structure of VA-I has long been sought after and will be well received by those in the field. The resulting structure shows an acutely bend RNA with several interesting features. The apical stem coaxially stacks with a four base pair stem near the central core. The central core features a compact pseudoknot structure that results in the acute bend between the apical stem and the truncated terminal stem. The manuscript examines the apical stem, tetrastem, and central domain structures and provides functional data to probe the importance of select interactions and features on PKR inhibition. The results suggest that the VA-I coaxial-stacked apical stem/tetrastem structure is responsible for PKR inhibition and the authors are able to design a fusion fragment that recapitulates wild-type inhibition. The manuscript represents a breakthrough in this field, and the structural data are of high quality. As such, the manuscript deserves publication in Nature Communications if the concerns below are addressed in a revised manuscript.

We thank the referee for her/his generous praise and favorable comments.

Major

1. The major shortcoming of the paper, which I believe can be easily remedied, is that the authors fail to present a cohesive story in relating the structural features of VA-I to their functions. The author's functional data mainly focuses on manipulating the apical stem length/sequence and shows that an apical/tetrastem fusion is capable of inhibiting PKR in the absence of the central domain. However they later offer hypotheses that suggest inhibitory functions for the central domain and the resulting bend. This contradicts their apical/tetrastem fusion RNA data that suggests a Wobble-rich stem, with no bend, is sufficient to inhibit PKR. They also suggest that the central domain may prevent PKR dimerization, however I do not believe that their functional data on the fusion RNA supports this hypothesis. The authors do say that their data supports a model in which the pseudoknot's main role is in stabilizing the coaxial stacking, but I feel it gets lost and minimized in the later statements. Simply cleaning up this region of text (Lines 307-320) and presenting a more cohesive story would greatly improve the clarity of the manuscript.

Following the referee's suggestions, we have reorganized the text in question (lines 307-320) and clarified the overall narrative. First we emphasize the statement that the pseudoknot likely aids folding and enables coaxial stacking.

“Based on our structure, the pseudoknot likely provides an anchor to position Stem 7, and topologically facilitates the coaxial stacking of the apical stem with the tetrastem, the key drivers of PKR inhibition.”

We reiterate this point later in Discussion.

“In contrast to its auxiliary role in PKR inhibition through facilitating RNA folding, the central domain pseudoknot is crucial for preventing PKR activation by a full-length VA-I.”

To further clarify the narrative, we consolidated all the text regarding the central domain's role in preventing PKR activation, moved them into Discussions, and organized them into three possible mechanisms. We now state in Discussions:

“There are at least three possible, mutually non-exclusive explanations for how the central domain prevents PKR activation. First, the central domain pseudoknot may function to restrain the overall shape or flexibility of VA-I to block PKR activation..... Second, the central domain pseudoknot may function as a steric barrier against deposition of a second PKR monomer, thus blocking PKR dimerization and activation on the same VA-I RNA..... Finally, the VA-I central domain may suppress PKR activation through direct contacts to the kinase domain or its adjacent basic patch in the inter-domain linker..... The precise mechanisms by which the VA-I central domain blocks PKR activation awaits further structural and biochemical analyses of PKR-VA-I complexes.”

2. The error bars for the inhibition data are not defined (standard error or standard deviation?). My eyes may be deceiving me, but some of the error bars on the inhibition plots for the mutants appear quite large and overlap with the error bars of the wild-type control. Because of this, readers cannot fully trust some of the trends seen. It would be good if the authors could collect additional replicates in order to make their functional data more robust. A few of the mutants that appear to have issues:

- a. Tetra Δ6 bp*
- b. Tetra Δ3 bp*
- c. AS only*

We thank for the referee for noting our oversight in the explicit definition of error bars. The error bars are standard deviations, not standard errors. We have now added an explicit notation in Fig. 1 caption.

“Error bars here and thereafter represent standard deviations (s.d.) from 3 independent replicates”.

Further, we have repeated and obtained more precise measurements for the 3 mutants in question, and updated the figures with the new data. Here is a comparison of the old and new data. The error spreads are much reduced, and the titration curves also agree with earlier data.

3. The authors give a single line suggesting crystal packing may have contributed to the differences between the solution SAXS and x-ray crystallography data. I believe that the authors should expand upon this and inform readers about this possibility and why it's so quickly dismissed as a possibility. The section on the “tRNA codon-anticodon mimicry” could be

replaced with a new small section in the manuscript that describes the crystal packing and implications. Many RNA crystal structures have exhibited crystal packing interactions, including stabilizing “kissing loop” interactions. Several figures (at least supplemental) on this should also be included.

We have now added a new Supplementary Fig. 14 to illustrate the detailed crystal-packing arrangements at three interfaces, and discussed in detail the potential influences on the overall and local structure. Now we state in Discussion:

*“There are at least two mutually non-exclusive possibilities for the differences. **First**, crystal packing could have altered the structure locally or globally. To understand the potential influence of the crystal-packing contacts, we analyzed the packing interfaces (Supplementary Fig. 14). The primary packing contact occurs at the top of Stem 7, where the single-stranded trinucleotide linker C116-G117-A118 pairs and stacks with its crystallographic symmetry mate. Interestingly, G117 bridges C116 and A118 of its symmetry mate forming a base triple and the two triples stack against each other, forming a presumably stable crystal-packing contact. As a previous SHAPE analysis confirmed the single-stranded nature of the trinucleotide linker, it seems unlikely that this packing contact drastically altered the overall structure. Nonetheless, the local structure of the trinucleotide linker that caps Stem 7 may have been locally deformed. A second packing contact occurs between the terminal G26 residue and its symmetry mate. This single, non-specific stacking interaction does not seem capable of drastically impacting the RNA conformation. Lastly, G127 intercalates between G64 and A66 of the apical loop. In this region, SHAPE analyses and nuclease sensitivities are consistent with the crystal structure. Further, given the numerous specific contacts within the central domain, it is likely that a drastic pivoting motion of the terminal stem would disrupt the pseudoknot, whose formation was well supported by multiple methods.”*

We have also removed the separate section on tRNA anticodon-codon mimicry and incorporated it into the central domain section, as suggested by the referee.

4. The authors should have a crisper discussion of how their structural data explains PKR inhibition. Previous work by Puglisi and Cole had showed that VAI binding blocks dimerization of PKR required for activation. The authors hint at this, but a clear section in the discussion is needed.

We have now included new SEC-MALS data that shows 1:1 binding of full-length VA-I and the Δ TS (crystallization construct) constructs to PKR in a new Supplemental Fig. 15. In the discussion, we now state the following regarding previous findings from Puglisi and Cole and cite appropriate references.

*“**Second**, the central domain pseudoknot may function as a steric barrier against deposition of a second PKR monomer, thus blocking PKR dimerization and activation on the same VA-I RNA. This explanation is supported by previous biophysical analyses of PKR-VA-I complexes using analytical ultracentrifugation³⁵ and dynamic light scattering^{34, 41}. We further confirmed that PKR binds 1:1 to full-length and Δ TS VA-I RNAs using SEC-MALS analyses (Supplementary Fig. 15).”*

Minor

1. The authors needlessly expand upon the “codon-anticodon” similarity of the central domain. This region of structure is indeed of interest, but this section feels more like an attempt to artificially increase the interest in the work. I would recommend that the “kissing” interaction should just be included within the section on the central domain rather than have its own section, and move the figure to the supplementary material.

We have now shortened this discussion and condensed it with the central domain section. We feel that the side-by-side comparisons might be of interest to the wider readership of the journal and propose to retain it as a main figure.

2. The writing, especially in the introduction, is hampered by the large amount of acronyms and the author’s tendency to list many examples. Making a statement and supporting it with citations would clean up the writing immensely and make it more accessible.

We have shortened the introduction and reduced unnecessary examples to streamline the narrative.

3. The writing also contains instances of superfluous wording. For example, phrases like “necessary and sufficient” (Line 105) can be simplified for clarity and directness. Just say that the RNA construct is derived from the Dicer-processed form and contains all the elements necessary for PKR inhibition.

We have removed the “necessary and sufficient” phrase and reworded as suggested. We now state:

“This RNA is derived from the Dicer-processed form and contains all the elements necessary for PKR inhibition (Fig. 1c; Δ T5 for deletion of terminal stem).”

4. Figure 2 a/b does not clearly illustrate (to me) how the current structure alleviates the old RNase data. A reader unfamiliar with the literature on how these conformers were stabilized will likely be confused. This may just require some clarity in writing and not a new figure.

We have revised the figure caption and main text to improve clarity.

5. Line 359: I believe the authors meant “(~60°)” and not “(~120°)”.

Yes. Corrected.

6. Line 419 about targeting a cavity in VA-I feels very forced and another attempt to artificially increase the impact of the paper. This is the very first instance that this structural feature is described (in the discussion!). This feature should either be expanded on (with implications, comparisons with other RNAs, etc) or the comment about it removed.

We agree that this is a minor point and have removed this comment.

Reviewer #2 (Remarks to the Author):

The authors present a novel high-resolution (2.7Å) structure of the adenovirus Virus-Associated RNA VA-I. This model helps to explain previously ambiguous experimental data but it also has notable differences from previously reported models, particularly from a model generated through SAXS. The authors use this new data to explain and to further explore the functional determinants of the molecule's inhibitory activity against a particular protein target, PKR. However it is also clear that the activity of VA-I relative to PKR is not the only significant selection pressure to evolve the VA-I's particular overall structure as they describe it (A more thorough investigation of these other pressures may be beyond the scope of the manuscript). The authors use structural homology and present limited sequence alignment data to hint at the evolutionary history of the molecule. The authors touch upon interesting synthetic biology applications that can be further considered as a result of this work. Given that there are relatively few high-resolution structures of viral subdomains and that this work yields insight into viral innate immune system evasion, the work is broadly important in my opinion.

We thank the referee for his/her favorable assessments.

General and specific comments follow:

About PKR binding to VA-I domain: The authors show the impact of mutations/deletions on PKR inhibition/activation. It would have been informative to have information on PKR binding as well.

A survey of extensive previous work showed that binding is required for PKR inhibition, but the strength of binding does not generally correlate with PKR-inhibition potency (McKenna, S. A., Kim, I., Liu, C. W. & Puglisi, J. D. *Uncoupling of RNA binding and PKR kinase activation by viral inhibitor RNAs. Journal of Molecular Biology* 358, 1270–1285 (2006)). This was recently re-affirmed in Dzananovic, E. et al. *Impact of the structural integrity of the three-way junction of adenovirus VAI RNA on PKR inhibition. PLoS ONE* 12, e0186849 (2017), where no binding defects were observed for a panel of mutant VA-Is that exhibited a full spectrum of activities on PKR. This was the reason that we did not focus on binding assessments. Nonetheless, we have now included new SEC-MALS data that show 1:1 binding of several VA-I constructs to PKR (Supplementary Fig. 15).

About VA-I and VA-II: The authors discussed in introduction that there are two categories of these RNAs, VA-I and VA-II. What are the similarities and differences in these RNAs? Does the structure of VA-I tell anything about VA-II? A comparative analysis would be interesting.

We have now added a sequence and structural alignment between Ad2 VA-I and VA-II, in Supplementary Fig. 9. Indeed, the VA-I crystal structure provided experimental constraints that can better guide a structural alignment of VA-I and VA-II. This improved alignment revealed interesting similarities and differences between the two RNAs. It was reported that Ad2 VA-II does not primarily target PKR, and makes a larger contribution than VA-I towards targeting Dicer and producing the so-called mivaRNAs. We believe the major interruptions in the apical

stem may be responsible to render VA-II inactive towards PKR. We further added a paragraph on VA-II at the end of the main text:

“The structural constraints derived from the VA-I crystal structure allowed more accurate structural alignment of VA-I and VA-II using Dynalign II (Supplementary Fig. 9). VA-II can assume an overall secondary structure very similar to VA-I, albeit with major interruptions to its apical stem. VA-II also presumably lacks the central domain pseudoknot. Its “tetrastem” appears to be 5 bp in length with essentially the same sequence as in VA-I. The interruptions in the apical stem resemble the AS bulge insertions that reduced PKR inhibition (Supplementary Fig. 9) and are probably responsible for VA-II’s minimal PKR-inhibitory activity observed in vivo 58. Instead of targeting PKR, VA-II which originated from a gene-duplication event from VA-I, is better suited to target other host enzymes such as Dicer5 and ADAR.”

About X-ray data and refinement: The Rmerge for the highest resolution cell is 364% and $I/\sigma(I)$ is 0.82. It means the meaningful resolution might not be 2.7-Å. What is CC1/2 for the overall data and for the highest resolution shell?

We have now included CC1/2 and CC* statistics for all the crystallographic data. The CC1/2 for overall and highest-res shell for the refinement dataset is 0.998 and 0.340, respectively. As a CC1/2 of 0.3 is approximately equivalent to $p = 0.01$ in statistics, we feel that this is a reasonable threshold to cut the data, in line with common practices in recent literature. We decided not to discard the weaker data in light of the discussions in the field about not truncating data to artificially improve Rfree. Kay Diederichs of the University of Konstanz has researched this topic extensively (Karplus, P. A. & Diederichs, K. (2012). *Science*, 336, 1030– 1033). She further commented that “*weak reflections (cutoff at $\langle I/\sigma \rangle \sim 0.5$ to 1, $cc1/2 \sim 0.15-0.25$) improved the model’s Rfree at lower resolution (paired refinement!) by a few percent, compared to cutting the data at $I/\sigma > 2$.*” <http://www.phenix-online.org/pipermail/phenixbb/2013-May/019791.html>

Line 183: Please comment on the conservation of the wobble pairs?

The wobble base pairs are indeed conserved. This is evident from Figures 6, 7, and 8 in Ma, Y. & Mathews, M. B. *Structure, function, and evolution of adenovirus-associated RNA: a phylogenetic approach. J. Virol.* 70, 5083–5099 (1996). In addition to Ad2 VA-I (this work), other adenoviral VA-I and VA-II from other serotypes (Ad4, Ad7, Ad12, Ad14, Ad15) also contain wobble pairs at similar locations. We now state this explicitly and refer to this paper and also refer readers to Supplementary Fig. 9 where VA-II also contains similar wobble pairs.

Line 190: Shouldn’t the sentence read “nonequivalence of binding and activation”?

Yes. Corrected.

Line 241: Mention that there could be as yet unascribed functions to the central domain

We actually briefly mentioned this at the end of the introduction.

“...and may hint at novel, undescribed functions of VA-I’s central domain.”

To reinforce this notion, we have now added another statement later at the end of the central domain Discussion:

“It is further possible that the VA-I central domain uses its unique 3D structure to execute pro-viral functions that have not been well-characterized, such as the inhibition of RNA helicases⁶⁹

Line 249: The authors state that “A123 employs a protonated N1 hydrogen bond to G106” but no protonation needs to be invoked. N3 can accept a hydrogen bond from the exocyclic amine of G106 and N1 can accept a hydrogen bond from G106 2'-hydroxyl group.

We thank the referee for pointing this out. Indeed protonation is not needed to engage the two hydrogen bonds involving N1 and N3 of A123 with G106. It is of interest how potential protonation of A123 at low pH might impact the strength of these two bonds. Careful pKa measurements would be necessary to investigate this further. We have revised this statement to say:

“Interestingly, A123 employs two hydrogen bonds from its N1 and N3 to G106’s 2'-OH and 2-exocyclic amine, respectively (Fig. 4b). These bonds likely underlie the pH sensitivity observed in thermal denaturation analyses, as the low-pH specific cooperative transition corresponding to central domain unfolding was eliminated by an A123U substitution”

Lines 269-277: Regarding Tb³⁺ cleavage, in my opinion this assay is a form of in line probing with Tb³⁺ acting as a catalyst to accelerate cleavage. In-line probing cleavage sites identify residues in the RNA that are flexible, so that the 2'-OH can get in line with the adjacent P-O5' bond. I disagree that it is simply a measure of solvent accessibility. Riboses in duplex regions are solvent accessible but are not good sites for Tb³⁺ cleavage for example.

We thank the referee for raising this important point. The referee is right that it is not accurate to state that Tb³⁺ sensitivity reports general solvent accessibility. It likely reports the accessibility of the region to Tb³⁺ in the solvent, the resident time for Tb³⁺ if bound here, and the probability of sampling the in-line configuration for cleavage, i.e., local backbone geometry and flexibility. We have revised the interpretation to state the following and modified Supplementary Fig. 12 to illustrate our proposal:

“We propose that the exceptional Tb³⁺ sensitivity of G97 and its opposing strand is primarily attributable to the unique geometric and electrostatic properties of its neighboring U96•G112 wobble base pair (Fig. 1b; Supplementary Fig. 7f). G•U or U•G wobble pairs drive a ~2Å outward shift of the uridine base into the major groove, forming a contiguous concave of electronegativity favorable for cation binding. In addition, the shifted U96 nucleobase is well situated to form a cation- π interaction to stabilize a Tb³⁺ bound to the N7 edge of G97 (Supplementary Fig. 12). Thirdly, the backbone geometry of wobble pairs exhibits higher plasticity and thus has a higher probability to engage the in-line configuration required for backbone scission. Similarly, the U45•G88 wobble pair region in the apical stem also exhibited strong cleavage both in the presence or absence of Tb³⁺ ions³¹. These findings illustrate a potentially general utility in using Tb³⁺ sensitivity to identify wobble base pairs in addition to locating specific Mg²⁺ sites in RNA^{47,48}. This is further consistent with preferred binding of Ir³⁺(NH₂)₆ to tandem G•U pairs that has been used as a general strategy to obtain phase information for RNA crystals.”

Line 307: This section is confusing. Above, the authors state that the central domain can be removed and show that the AS-Tetrastem fusion has inhibition properties that exceed the wild-type. Yet, here the authors try to ascribe a role for the central domain inhibition versus activation of PKR. What is the reason for this discrepancy? Does the role of the central domain become important only in nonfused tatrastem constructs?

That is exactly what we believe, that the central domain function primarily in the context of a longer RNA that normally runs the risk of activating PKR. The apical-tetrastem fusion RNA (mini-VA) is an artificial construct that is much shorter and dsRNA only. In this context the central domain is dispensable for PKR inhibition. We believe that VA-I evolved other functions that required additional sequences, such as its terminal domain that targets Dicer. The central domain pseudoknot and its acute bend may serve to suppress the PKR activation potential associated with the longer RNA platform and additional sequences. We have reorganized this section in Discussions to clarify this. Please also refer to our response to the first major comment from Referee #1.

Discussion: Perhaps state that it is possible that the domain has some flexibility with respect to being acutely bent versus more obtuse. Flexibility is frequently a functional feature of structured RNAs.

We agree and have added a discussion regarding this, immediately following the differences between X-ray and SAXS envelopes. We now state:

“In solution and without the constraints of the crystal-packing lattice, the VA-I RNA is likely more flexible and can sample additional conformations such as the more obtuse angles, or even transiently disengage its pseudoknot interaction. It is presently unknown if such structural flexibility is important for VA-I’s ability to target more than one host proteins.”

Figure 2: It would be best to make all of the y-axes the same for each of the inhibition plots. All the plots are now placed on the same y-axes between 0 and 1.2.

REVIEWERS' COMMENTS:

Reviewer #1 (Remarks to the Author):

The authors have addressed our concerns, and the revised manuscript is acceptable for publication.

Reviewer #2 (Remarks to the Author):

The authors have done a good job addressing the comments from the first round of review; they obtained binding data, did a bit more comparative analysis, cleared up some confusing points and touched on additional interpretations/possibilities.

This is a strong paper.

REVIEWERS' COMMENTS:

Reviewer #1 (Remarks to the Author):

The authors have addressed our concerns, and the revised manuscript is acceptable for publication.

Reviewer #2 (Remarks to the Author):

The authors have done a good job addressing the comments from the first round of review; they obtained binding data, did a bit more comparative analysis, cleared up some confusing points and touched on additional interpretations/possibilities.

This is a strong paper.

We thank both reviewers for their insightful and constructive comments, which have helped us improve the quality and clarity of our manuscript.